# Lab-in-Syringe, a Useful Technique for the Analysis and Detection of Pollutants of Emerging Concern in Environmental and Food Samples

**DOI:** 10.3390/molecules27217279

**Published:** 2022-10-26

**Authors:** Víctor Cerdà, Sergio L. C. Ferreira, Piyawan Phansi

**Affiliations:** 1Sciware Systems, S.L., 07193 Bunyola, Spain; 2Instituto de Química, Universidade Federal da Bahia, Bahia, Salvador 40170-270, Brazil; 3Department of Chemistry, Thepsatri Rajabhat University, Lopburi 15000, Thailand

**Keywords:** lab-in-syringe, dispersive liquid-liquid microextraction, sample treatment automation

## Abstract

Lab-in-syringe is a new approach for the integration of various analytical extraction steps inside a syringe. Fully automated dispersive liquid–liquid microextraction is carried out in-syringe using a very simple instrumental setup. Dispersion is achieved by aspiration of the organic phase and then the watery phase into the syringe as rapidly as possible. After aggregation of the solvent droplets, the organic phase is pushed towards the detector allowing a highly sensitive spectrophotometric or fluorimetric detection. This technique is very useful not only for the preconcentration of analyte, but also for the elimination of their interferences. In this work, its application is described using solvents that are lighter and denser than water. The magnetically assisted variant and its coupling to different instruments has been also described with the aim of increasing the resolution of complex samples, especially useful for the determination of emerging contaminants.

## 1. Introduction

The Dispersive Liquid-Liquid Microextraction (DLLME) technique [1] was developed by Rezaee et al. [2]. DLLME is a fast microextraction technique based on the use of a small amount of an appropriate immiscible extractant solvent and a ternary component, which is named as the disperser solvent.

In this technique, the disperser solvent is miscible both in water and with the extractant solvent. The extractant and the disperser are mixed and injected rapidly into the sample to produce a turbulent mixture due to the formation of small droplets of the extractant throughout the aqueous sample. The formation of the small droplets enhances the effective surface area of the liquid-liquid extraction, being the equilibrium reached in a shorter time. In the manual procedure, the two phases are separated usually by centrifugation, being a small volume of the extractant settled at the bottom of centrifugation tube.

The target compounds extracted in the recovered phase after DLLME are usually organic compounds or metallic complexes, which are subsequently analyzed using different techniques such as High Performance Liquid Chromatography (HPLC) [3,4], Gas Chromatography-Electron Capture Detection (GC-ECD) [5] or Gas Chromatography-Mass Spectrometry (GC/MS) [6].

One of the main limitations of the DLLME technique was its automation [7]. Some advances were accomplished using the Sequential Injection Analysis technique (SIA) [8,9] in combination with an additional peristaltic pump, mixing by confluence the sample and the disperser/extractant mixture [10,11]. However, a micro-column packed with polytetrafluoroethylene (PTFE) was required as a solid support for the retention/collection of the dispersed organic micro-droplets and an additional organic solvent was required for the elution of the retained droplets in the PTFE support prior to the determination of the analytes by means of atomic absorption spectrometric techniques.

The lab-in-syringe (LIS) is a new technique which has proven to be very powerful for pre-concentration, cleaning and elimination of interferences in the determination of very different types of samples, both organic and inorganic. It can be applied in liquid-liquid or liquid-solid extraction with good results. Its coupling with great instrumentation is especially interesting, such as chromatographic techniques and atomic spectroscopy, which provide a great selectivity and sensitivity in the determinations. Therefore, LIS may be very useful for the determination of emerging pollutants both in environmental and food samples at very low determination limits.

The aim of the present work is to revise the Lab-in-syringe technique (LIS) for the complete automation of DLLME based on its original concept, and without the requirement of additional steps, enabling the direct injection of the separated extractant into hyphenated analytical instrumentation in a simple and quick fashion [12].

## 2. Results

### 2.1. Lab-in-Syringe for Extracting Solvent Lighter than Water

The first attempt to automate the DLLME technique using a flow system was due to F.Maya et al. [13]. The developed system is shown in Figure 1a. DLLME is carried out In-Syringe using solvents lighter than water. In this case, 1-octanol was selected as a model extractant.

Syringe 1 (S1, 5 mL) was used for the automated In-Syringe DLLME. S1 was connected to the central port of a rotary selection valve (SV), enabling the aspiration of the extractant/disperser mixture, and immediately after, the sample was loaded at a high flow rate (15 mL min^−1^) being produced a cloudy solution inside the syringe. A dead volume of approximately 150 µL (between the head of S1 and the central port of the SV) prevented the contact between the extractant/disperser mixture and the syringe plug minimizing the carry over.

After a short reaction time of 30 s the 1-octanol was separated from the mixture at the head of the syringe. S1 was connected to a water reservoir, which was used for the automated cleaning of the syringe between runs.

Once DLLME had been carried out, the versatility of the MSFIA technique was exploited for the determination of the analytes implementing a hyphenated liquid chromatography procedure in the same multisyringe pump, which is known as Multisyringe Chromatography (MSC) [14,15]. MSC is based on the implementation of short silica-C18 monolithic columns using syringe-based low-pressure flow systems, in this case using a multisyringe pump.

In order to combine the DLLME with the separation and detection steps, the extractant phase located in the head of the syringe was injected through the position 4 of the SV. It was then mixed with the solution contained in Syringe 2 (S2, 10 mL, ACN:H_2_O 90/10 *v/v*) in order to decrease the viscosity of the extractant and enable its injection through a chromatographic column. A 100 cm coil was used for the mixing of the extractant with the solution contained in S2.

An injection valve (IV) was used for the introduction of the extractant into the MSC system. The injection loop was filled with the diluted extractant. Changing the position of the IV the extractant was injected into the MSC system using the Syringe 3 (S3, 5 mL) containing the eluent. Chromatographic separation was achieved using a silica-C18 monolithic column (with a guard column), being the analytes detected by UV spectrophotometry using a long path length liquid waveguide capillary cell (LWCC) as a front end of the liquid chromatographic system [16,17].

This automated procedure solved one of the main drawbacks of DLLME using solvents less dense than water, which was the recovery of the extractant solvent. In this case the extractant was automatically managed being feasible their injection in separation techniques similar to the MSC system developed in this work.

The extractant solution located in the port number 3 of the selection valve included the disperser at a ratio of a 10% of extractant (1-octanol) and a 90% of disperser (acetonitrile, acetone or methanol). With the three different disperser agents, a cloudy solution was produced after the aspiration of the sample. For the extraction of 2 mg L^−1^ of naproxen (detected at 310 nm), best results were obtained using ACN. A standard deviation (RSD, %) for three consecutive extractions was a 2.8%.

The extraction time was studied, and best results were obtained with 30 s. Among the different size volumes of the glass syringes available for the multisyringe pump (0.5–10 mL), a syringe of 5 mL was selected in order to load the maximum volume of sample. Using syringes with a bigger volume (10 mL), a part of the 1-octanol was stacked at the head of the syringe, due to its wider shape.

In order to perform the determination of benzo(a)pyrene without syringe re-filling, the content of ACN in the mobile phase was increased up to a 70%, decreasing the retention time of benzo(a)pyrene. As is shown in Figure 2b, three replicates of the automated DLLME of benzo(a)pyrene (50 µg L^−1^) can be achieved with a RSD of the signal peak height of only a 1.4%.

The response of the DLLME of benzo(a)pyrene showed a good linearity in a range of concentration of 5–50 µg L^−1^ (slope = 0.0137 L µg^−1^, R2 = 0.9998). Using this configuration, a limit of detection (LOD) of 0.7 µg L^−1^ and limit of quantification (LOQ) of 2.4 µg L^−1^ were obtained for the measurement of benzo(a)pyrene followed at its excitation wavelength (295 nm), using the signal-to-noise ratio (S/N) criteria of 3 and 10. The usual working range using low-pressure chromatography based on short monolithic columns are on the mg L^−1^ level.

Figure 2c shows the difference between chromatograms obtained by direct injection of tap water spiked with 15 µg L^−1^ of benzo(a)pyrene without and with DLLME. We can also notice the similarity of the injection of spiked tap water using DLLME and the injection of a standard solution spiked with an equal amount of benzo(a)pyrene, demonstrating the effectiveness of the proposed DLLME procedure for the removal of undesirable sample matrix effects. The recovery for this sample was a +109%, and the increase on the measured absorbance due to the DLLME treatment was 12.5-fold.

### 2.2. Lab in a Syringe-Fully Automated Dispersive Liquid-Liquid Microextraction with Integrated Spectrophotometric Detection

A new approach for the integration of various analytical steps inside a syringe (Lab in a Syringe) was later presented [18]. Fully automated dispersive liquid-liquid microextraction (DLLME) with integrated spectrophotometric detection was carried out in-syringe using a very simple instrumental set-up (Figure 3). The organic droplets lighter than water released in the extraction step, accumulate at the head of the syringe, where two optical fibers are placed on both sides of the syringe, facing each other and enabling the in-situ quantification of the extracted compounds. By this, monitoring of the progressively accumulating droplet in the head of the syringe was further possible.

The developed instrumental set-up was applied to the determination of rhodamine B in water samples and soft drinks. The main parameters influencing the extraction such as the selection of the extractant and disperser solvents, extractant/disperser and organic/water phase ratios, pH of the aqueous phase, extraction flow rates, and extraction time were investigated. Under the selected conditions, rhodamine B was quantified in a working range of 0.023–2 mg L^−1^ with a limit of detection of 0.007 mg L^−1^. Good repeatability values of up to a 3.2% (RSD) were obtained for 10 consecutive extractions. The enrichment factor for a 1 mg L^−1^ rhodamine B standard was 23, and up to 51 extractions were accomplished in 1 h.

### 2.3. Lab-in-Syringe with Magnetic Stirring Assisted Dispersive Liquid–Liquid Microextraction (MSA-DLLME)

The original methodology requires a dispersion solvent as major component of the organic phase, which dissolves preferably in the aqueous phase at the rapid injection of the solvent mixture into the aqueous phase. Thus, a very small amount of extraction solvent is effectively dispersed into droplets, which afterwards are forced to coalesce by a centrifugation step. The organic phase is then transferred into the detector or used for chromatographic separation.

However, the dispersion solvent assisted DLLME has a few inconveniences. The additional solvent leads to increase waste production. The method requires additional optimization effort (dispersion solvent quantity and kind) and, the most important, the dispersion solvent increases the solubility of the analyte in the aqueous phase. Furthermore, the distribution of the dispersion solvent between both phases and by this, the final volume of the organic phase depends on the sample salinity.

Consequently, alternative DLLME methodologies have been developed, where extraction solvent dispersion is achieved by kinetic energy. Depending on the mode of achieving droplet formation or stabilization of the droplets in the aqueous phase, ultrasound-assisted DLLME [19], air-assisted DLLME [20], vortex-assisted DLLME [21], magnetic stirring-assisted DLLME (MSA-DLLME) [22], and surfactant assisted DLLME [23] can be distinguished among others. For details, the reader is referred to recent and extensive review articles on this topic [7,24,25,26].

#### 2.3.1. In-Syringe Magnetic Stirring Assisted Using an Extraction Solvent Lighter than Water

A novel approach using a magnetic stirring bar within the syringe pump of a SIA system was developed [27]. A sealed but adaptable reaction vessel was obtained, in which all solutions could be aspirated with high precision and mixed homogeneously and nearly instantaneous. If air and an extraction solvent lighter than water were used, vortex formation allows the contact of the extraction solvent with the turning stirring bar and hereby, the dispersion of the solvent into fine droplets. Stopping the stirring allows then droplet floatation, coalescence, and expulsion of the extraction solvent into a detection flow cell.

The system was used for the extraction of aluminum (Al^3+^) as lumogallion (LMG) complex from seawater samples. This also allowed a critical comparison with a similar application but based on in-syringe dispersion solvent-assisted DLLME [28]. In both works, LMG was chosen as a very selective fluorescence reagent for aluminum [29]. In contrast to the often-used morin, the LMG-Al complex is extractable into moderately hydrophobic organic solvents. The in-syringe magnetic assisted dispersive liquid-liquid microextraction (MSA-DLLME) manifold is depicted in Figure 4a. The computer-controlled flow setup comprised a multisyringe pump and the rotary 8-port selection valve (SV) for liquid handling and distribution. The multisyringe pump was equipped with a sole glass syringe (S) of 5 mL. A three-way solenoid head valve (V) on-top of the syringe enabled the connection to either the central port of the SV (position ON, activated) or to the detection cell and downstream located waste for quantification of the extracted analyte as well as for discharge during syringe cleaning (position OFF, deactivated).

Peripheral ports of SV were connected to reservoirs of waste (1), water (2), sample (3), buffer (4), lumogallion reagent (5), n-hexanol (6), air (7), and acetonitrile (8). Water and acetonitrile were used for cleaning of the detection flow cell or the syringe, which was routinely carried out daily.

The connection between the central port of the SV and the syringe head valve was carried out by a short holding coil (HC). Heating was carried out to favor the slow reaction between LMG and Al^3+^.

The entire analytical procedure was carried out in the syringe including sample mixing with reagents and extraction. To achieve homogeneous and rapid mixing without an additional mixing chamber as generally carried out [28,30,31], a magnetic Micro stirring bar (10 mm length, 3 mm diameter) was used within the syringe. The top position of the syringe piston was adjusted in such a way, that a gap less about 0.5 mm was left at emptying the syringe to avoid any damage.

To drive the stirring bar in the syringe, a commercial magnetic laboratory stirrer was impractical. Therefore, a rotating magnetic field was achieved by the use of a specially developed magnet driver, shown in Figure 4b. Two rings made of nylon were used as bearings, which could be placed easily onto the syringe, with the bottom ring sliding on the flange of the syringe barrel. Two metric 4 steel screws of 80 mm in length were used as spacers and connection between both nylon rings. The so-obtained assembly could freely rotate around the syringe longitudinal axis.

By placing two neodymium magnets on top of the screws, the screws were magnetized and thus, a magnetic field in the syringe along its whole length was obtained. This magnetic force was sufficient to attract and, at turning the device to force the rotation of the stirring bar inside the syringe independently from the position of the syringe piston.

The bottom ring showed further a groove for the placement of a rubber band, which allowed propelling the driver with a low-cost DC motor. The DC motor was activated using a homemade relay and regulation circuit board by an auxiliary supply port of the multisyringe module.

A specially made detection cell was used for fluorescence measurements. A detailed description of the cell design can be found elsewhere [28]. It comprised a glass tube of 3 mm id used as detection cell flow channel. A bright green LED with an emission wavelength of 500 nm, powered by a mobile phone charger, was used as excitation light source and aligned with the glass tube. A photomultiplier tube (PMT) from Hamamatsu Phototonics K.K. (Hamamatsu, Japan, Ref.: HS5784-04) was used for detection of fluorescence emission and was mounted in perpendicular position onto the glass tube.

In addition, a polycarbonate collector lens (F 22 mm, O 22 mm) was placed onto the PMT to achieve higher sensitivity. A control unit from Sciware Systems, S.L. (Bunyola, Spain) was used for PMT supply and data readout. A gain of 18% was chosen for the PMT.

The software AutoAnalysis 5.0 (Sciware Systems, S.L., Bunyola, Spain) was used for operational control of the flow instrumentation as well as data acquisition from the detection equipment and data evaluation.

The operation methods for testing in-syringe dilution and homogenization as well as MSA-DLLME are given schematically in Figure 5.

All analytical procedures required the cleaning of the syringe due to the unavoidable dead volume caused by the stirring bar. It was given by the syringe inner diameter of 10.5 mm and the height of the magnetic stirring bar of 3 mm minus its proper volumetric displacement of about 70 μL. However, the cleaning could be performed efficiently because the stirring allowed instantaneous homogenization of the dead volume in the syringe with the cleaning solution. Three-fold aspiration of 0.8 mL of water (V in position ON, stirring activated) and discharge to waste (V in position OFF) was sufficient and allowed syringe cleaning in less than 30 s. In addition, procedures for cleaning of supply tubes on the SV and the detection cell were established.

In-syringe dilution and homogenization were studied using 1 mg L^−1^ rhodamine B solution and fluorimetric detection. Subsequently, 1 mL of rhodamine solution, 3 mL of ultrapure water, and 200 μL of air were aspirated into the syringe omitting stirring. Next, the syringe content was mixed by activation of the stirring for a defined time. Afterwards, the stirring was stopped, and the syringe content was dispensed through the detection cell for the evaluation of the achieved homogenization.

MSA-DLLME was started by the aspiration of 240 μL buffer, 60 μL of LMG reagent, and 4.1 mL of sample into the syringe. Sample aspiration was carried out at a reduced flow rate of 4 mL min^−1^ to increase the heat transfer from the heating device to the sample and during repeated activation of the in-syringe stirring. Next, the stirring was deactivated and during a reaction time of 15 s, 150 μL of n-hexanol were aspirated into the HC to heating it up.

Afterwards, the stirring was started again and 400 μL of air were aspirated so that the n-hexanol in the HC and also part of the air could enter the syringe. The air allowed the formation of a vortex in the syringe. At contact of the organic phase with the stirring bar, it was dispersed into small droplets. The stirring was kept activated for 40 s to perform MSA-DLLME. The stirring speed was 2000 min^−1^.

The stirring was stopped, which allowed the formed n-hexanol droplets to float and coalesce during 30 s at the brim of the concave liquid meniscus formed by the aqueous phase in the syringe. To improve droplet aggregation, the liquid surface was put in movement by a short movement of the piston (approx. 1 mm) by the instruction of complete filling just before the next step. The method was finalized by pushing the organic solvent, enriched with the LMG-Al complex, slowly through the detection cell to waste under continuous data evaluation. Finally, the remaining liquid was rapidly discharged from the syringe to waste.

The optimized method enabled efficient DLLME within a comparably short time and is based on the disruption of the extraction solvent by the kinetic energy of the swirling stirring bar. Better or similar analytical performance than in previous works based on DLLME was achieved and the method’s applicability to the determination of aluminum in surface seawater and freshwater samples was proven (see Table 1). Dependency of the analytical performance on the sample salinity and viscosity was demonstrated to be widely overcome. In-syringe stirring can enable novel protocols for sample preparation, analyte pre-concentration, and complex analytical applications.

#### 2.3.2. In-Syringe Magnetic 1 Stirring Assisted Using an Extraction Solvent Denser than Water

An automated simple analyzer system for the extraction of cationic surfactants as an ion-pair with disulfine blue dye has been described based on the technique in-syringe magnetic stirring-assisted dispersive liquid–liquid micro-extraction [37]. The use of chloroform as an extraction solvent denser than water required the operation of the syringe pump upside-down. The remaining air cushion inside the syringe allowed emptying the syringe completely and reducing the dead volume significantly. Since the stirring bar placed inside the syringe to obtain a closed yet size-adaptable mixing chamber remains at the same position, the former magnetic stirring bar driver was simplified (Figure 6). The position of the syringe piston was adjusted to leave a gap of 4 mm on complete emptying, so that the stirring bar could freely rotate.

The operation scheme is given in Figure 7. The procedure started with the cleaning of the syringe by threefold aspiration of 0.6 mL of the sample or the respective standard solution from the SV under high-speed stirring and dispensed through the head valve position OFF to waste.

Next, buffer, DSB solution, and the sample were aspirated into the syringe under low speed stirring for homogenization. The required volume of the organic phase was aspirated followed by a volume of air being large enough to fill the HC, so that the organic phase entered the syringe completely. High speed stirring was carried out for 35 s for DLLME. Here, it was found advantageous to start and end with 5 s of stirring at lower speed to overcome the inertia of the solution at starting and to improve posterior droplet coalescence, respectively.

After phase separation and droplet coalescence, either the organic phase was pushed slowly through the detection cell followed by emptying the syringe completely at a high speed (procedure 1) or, for extract washing, the organic phase was pushed into the HC, and then, the remaining liquid was dispensed through the detection cell to waste (procedure 2). In procedure 2, the extract was re-aspirated into the syringe together with water, barium acetate, and DSB solution, followed by another DLLME step, phase separation, and then measurement. An additional washing step with pure water was carried out as before performing the extraction step with barium acetate. A 40 mL larger volume of the organic solvent was required for procedure 2 since a part of the organic phase would dissolve in the aqueous sample and washing solutions.

High signal repeatability with <5% RSD was achieved both for extraction as well as for double organic phase washing. Only 220 mL of the extraction solvent and 4 mL of the sample were required for simple extraction achieving a detection limit below 30 nmol L^−1^ and a linear response up to 1 mmol L^−1^ of cetyltrimethylammonium bromide. The time of analysis was 240 s for simple extraction. Considerable reduction of interference was achieved by extract washing up to 545 s. Analyte recovery in real water samples was 95.6 ± 7.0% on applying extract washing.

### 2.4. In-Syringe Dispersive Microsolid Phase Extraction Using Magnetic-Metal Organic Frameworks

An automatic strategy for the use of micro and nanomaterials as sorbents for dispersive microsolid phase extraction (D-μ-SPE) based on the lab-in-syringe concept has been reported [38].

Using the developed technique, the implementation of magnetic metal-organic framework (MOF) materials for automatic solid-phase extraction was achieved for the first time.

A hybrid material based on sub-micrometric MOF crystals containing Fe_3_O_4_ nanoparticles was prepared and retained in the surface of a miniature magnetic bar. The magnetic bar was placed inside the syringe of an automatic bidirectional syringe pump, enabling dispersion and subsequent magnetic retrieval of the MOF hybrid material by automatic activation/deactivation of magnetic stirring (See Figure 8).

Using malachite green (MG) as a model adsorption analyte, a limit of detection of 0.012 mg L^−1^ and a linear working range of 0.04–2 mg L^−1^ were obtained for a sample volume equal to the syringe volume (5 mL). MG preconcentration was linear up to a volume of 40 mL, obtaining an enrichment factor of 120. The analysis throughput is 18 h^−1^, and up to 3000 extractions g of material can be performed. Recoveries ranging between 95 and 107% were obtained for the analysis of MG in different types of water and trout fish samples. The developed automatic D-μ-SPE technique is a safe alternative for the use of small-sized materials for sample preparation and is readily implementable to other magnetic materials independent of their size and shape and can be easily hyphenated to the majority of detectors and separation techniques.

A similar procedure was developed using a commercial SupelTM-Select HLB (Hydrophilic modified styrene polymer) sorbent beads embedding magnetite nanoparticles (Fe_3_O_4_) [39]. The sorbent was then used in a dispersive solid phase extraction procedure that was carried out entirely inside the void of an automatic syringe pump following the flow-batch concept of Lab-In-Syringe including automated renewal of the sorbent for each analysis. Mixing processes, sorbent dispersion, and sorbent recovery were enabled by using a strong magnetic stirring bar, fabricated from a 3D printed polypropylene casing and neodymium magnets, inside the syringe. The final extract was submitted to online coupled liquid chromatography with spectrometric detection. System and methodology were applied to determine mebendazole, bisphenol A, benzyl 4-hydroxybenzoate, diclofenac, and triclosan selected as models from different groups of environmental contaminants of current concern.

### 2.5. Hyphenation Lab-in-Syringe with Big Instruments

Quite often one of the biggest problems in analytical methods resides in the pre-processing of the sample, which usually must be solved manually. To solve these problems, it can be very useful to request the help of the flow techniques, including LIS, which will allow for the automation of the methods, decrease the time of the analysis, decrease the consumption of the samples and reagents, increase the reproducibility, the sample throughput and to work continuously for a long period of time.

Between the advantages of the separation flow techniques are their very high versatility facing the application of various treatment techniques, such as photo-oxidation, pre-concentration, clean-up, derivatization, gas diffusion, etc. Since the entire process is carried out in a completely closed system, results are obtained in a faster and more reproducible way.

However, although the LIS can solve the selectivity problem of simple sample determinations, it is insufficient for the resolution of relatively complicated samples. Therefore, there is no other alternative than to couple it with other techniques with a higher resolution capacity, such as such as chromatographic techniques for the analysis of organic compounds, or atomic spectroscopy for the determination of inorganic species [40].

In this contribution, coupling LIS with several very selective spectrometric instruments, such as ICP–AES, ICP–MS, AFS, etc. Hyphenation with different kind of chromatographic and the capillary electrophoretic techniques is also described.

#### How to Couple Flow Techniques

There are several ways to hyphenate flow techniques. Usually, it is necessary to use two personal computers (PC), one controlling the low-resolution flow technique, which through the “events input”, will trigger the program placed in the other PC controlling the high-resolution task connection.

Multisyringe Flow Injection Analysis (MSFIA) is a very well adapted to do this task by means LIS, using a multisyringe. Figure 9 show how to hyphenate LIS with atomic techniques and anodic stripping voltammetry, and high resolution chromatographies.

When using a multisyringe burette, very important components for the hyphenation tasks are the four ports placed on the rear panel of the burette (Figure 10), which enable to power of external multi-commutation valves, micropumps or other instruments either directly or via a relay allowing remote software control (e.g., ICP-MS). This amplifies the possibilities to construct sophisticated flow networks.

The procedure to operate with this hyphenated system is the following:Prepare the high-resolution instrument to do its task using computer 2, but leaving it in standby waiting to be triggered by the multisyringe.Use computer 1 with the aid of the program AutoAnalysis [41] to perform all pre-treatment tasks by means of the LIS system, such as pre-concentration, clean-up, derivatization, etc.Once finished the pre-treatment tasks, trigger the computer 2 of the high-resolution instrument in order to run its program and make the analytical measures.Whereas the high-resolution instrument is working, the LIS system works simultaneously preparing the next sample using computer 1.

### 2.6. Hyphenated Technique of LIS with Chromatographic Techniques

One example is the determination of UV filters by chromatographic techniques [42]. Despite of the fact that gas chromatography (GC) presents higher resolution, offers good separation and lower limits of detection (LOD) for the environmental analysis of UV filters [43,44], high performance liquid chromatography (HPLC) [45,46] is a good option to determine UV filters since it is a more affordable technique and good performance is accomplished. Furthermore, due to the low volatility character of UV filters, HPLC is preferred because analyte derivatisation is avoided.

A work was developed [47] as a fast, simple, fully automated, cost-effective and environmentally friendly method based on in-syringe MSA-DLLME coupled to HPLC allowing the on-line extraction, preconcentration, separation and detection of six UV filters in water samples (Figure 11). The main advantage of the method relies in the use an on-line sample treatment through the in-syringe MSADLLME system, exploiting an ion liquid as extractive solvent instead of chlorinated solvents, becoming a greener alternative to existing methods.

Under optimized conditions, detection limits were within the range of 0.08–12 mg L^−1^, for 3.5 mL sample volume. Linearity ranges were up to 500 mg L^−1^ for the UV-filters studied. Furthermore, enrichment factors ranging from 11 to 23 folds were obtained. Intra- and inter-assay precisions were 6% and 8%, respectively. The proposed method was successfully applied to determine UV filters in surface seawater and swimming pool samples attaining satisfactory recoveries over the range of 89–114% and 86–107%, respectively.

Figure 12 shows typical chromatograms of raw seawater and spiked seawater analyzed with the proposed system. As can be seen under optimized chromatographic conditions, the peaks were well resolved and endogenous environmental water compounds did not give any interfering peaks.

Another example was the use of a hyphenated LIS-GC/MS system applied to the determination of six phtalates (PAEs) in waters (Figure 13) [48].

To investigate the effect of sample matrices on extraction efficiency, these samples were spiked with the target compounds at 5 μg L^−1^ concentration level. Relative recoveries of the PAEs at two different spiked concentrations were satisfactory, i.e., in the range of 82 and 111%. This proved that the developed procedure is suitable for genuine environmental and consuming water applications.

Figure 14 shows the chromatograms of a tap water sample extract and its spiked standard solution analyzed with the proposed system.

A dispersive liquid–liquid microextraction (DLLME) method was applied using high performance liquid chromatography (HPLC) to the determination of 15 polycyclic aromatic hydrocarbons (PAHs) (Table 2) in aqueous matrices (Figure 15). The extraction procedure was automated using a system of multisyringe flow injection analysis coupled to an HPLC instrument provided with a fluorescence detector.

The MSFIA–DLLME–HPLC proposed method was applied for the determination of 15 priority PAHs in water samples to demonstrate the applicability and reliability of the method. Three different kinds of samples: tap water, rainwater and stream surface water were analyzed. To estimate the matrix effect, all the samples were spiked with 0.3 μg L^−1^ of PAH individual standards to calculate the recovery of the targeted compounds. The sample results and the recovery study were performed in triplicate (see Table 4). No PAHs were found in the analysis of tap water samples. However, some PAHs were found in rain or stream water samples. Only PAHs of 2 or 3 rings were detected. This is due to their solubility in water, which decreases with the increase of molecular weight. Five or six ring PAHs in aqueous medium were only found in highly contaminated sites.

## 3. Conclusions

The lab-in-syringe (LIS) is a new technique which provides several interesting performances:-Has proven to be a very powerful for pre-concentration, cleaning and elimination of interferences in the determination of very different types of samples, both organic and inorganic.-It can be applied in liquid-liquid or liquid-solid extraction with good results.-May be coupled with great instrumentation is especially interesting, such as chromatographic techniques (HPLC, MSC, CG and GC/MS) and atomic spectroscopy (graphic camera, ICP_OES, ICP-MS, atomic fluorescence), which give it great selectivity and sensitivity in the determinations.-Therefore, LIS may be very useful for the determination of emerging pollutants both in environmental and food samples at very low determination limits.

## Figures and Tables

**Figure 1 molecules-27-07279-f001:**
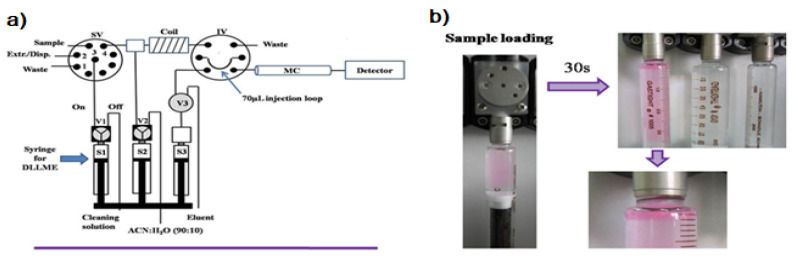
(**a**) Developed instrumental set-up for DLLME combined multisyringe chromatography (MSC) [13]: S1–S3, syringes. V1–V3, solenoid valves. SV, 8-port selection valve. Extr./Disp., extractant/disperser mixture. IV, injection valve. MC, monolithic column. (**b**) Sample aspiration through the extractant/disperser producing a cloudy solution and phase separation after a reaction time of 30 s. Rhodamine B was used as the colored product.

**Figure 2 molecules-27-07279-f002:**
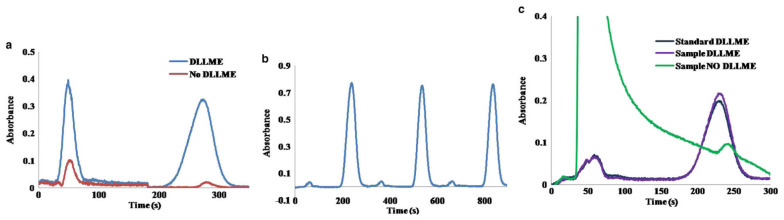
(**a**) Chromatogram showing the separation of benzo(a)pyrene (25 μg L^−1^, 273 s) from the phenol (50 s, 1 mg L^−1^) without and with DLLME; (**b**) Three replicates of the injection of a 50 μg L^−1^ benzo(a)pyrene standard using ACN/H_2_O 70/30 *v*/*v*; (**c**) Chromatograms corresponding to the direct injection of tap water spiked with 15 μg L^−1^ benzo(a)pyrene spiked tap water after DLLME, and a standard containing 15 μg L^−1^ of benzo(a)pyrene using DLLME. Mobile phase, ACN/H_2_O (70/30 *v*/*v*) [13].

**Figure 3 molecules-27-07279-f003:**
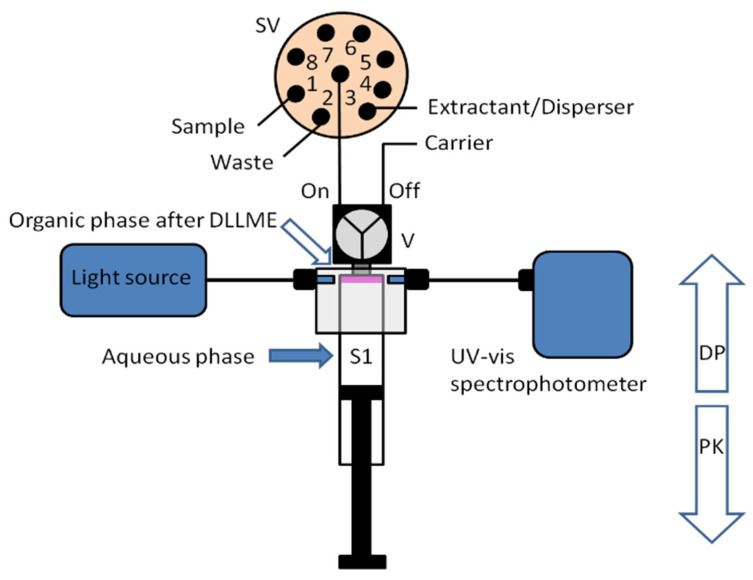
Schematic representation of the developed system for in-syringe DLLME with integrated spectrophotometric detection [18].

**Figure 4 molecules-27-07279-f004:**
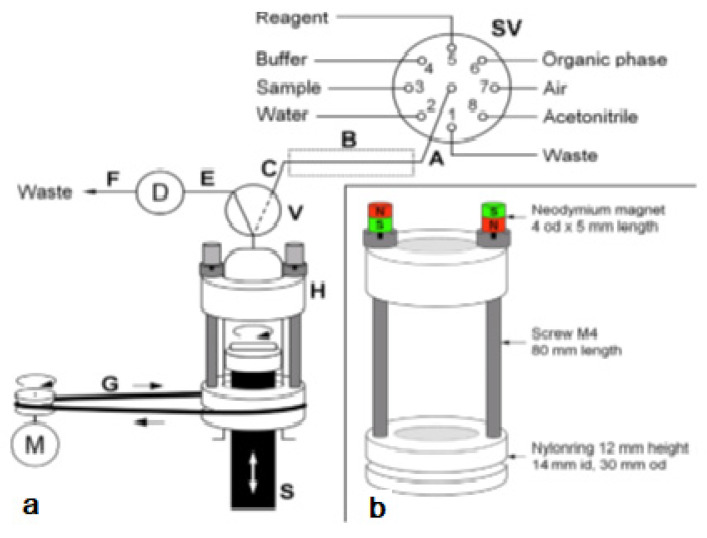
(**a**) Analyzer manifold with selection valve (SV), syringe pump (S), solenoid 3-way head valve (V), detection flow cell (D), heating device integrated into the HC (B) and the magnetic stirring bar driver (H) placed onto the syringe barrel. A motor (M) is used to drive it via a rubber band (G). PTFE tubing (0.8 mm i.d.) of 15 cm (A,C), 10 cm (E), and 40 cm (F): 10 cm.; (**b**) The magnetic stirring bar driver placed onto the syringe glass barrel shown in detail [27].

**Figure 5 molecules-27-07279-f005:**
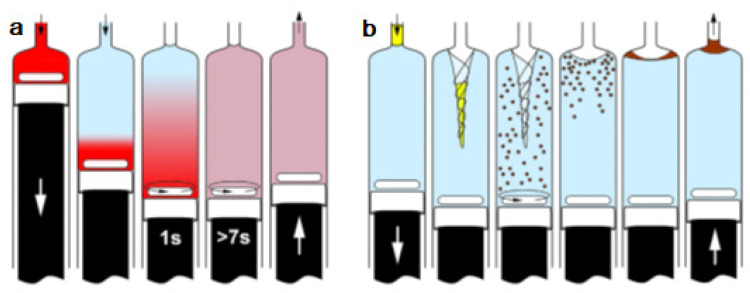
Schemes of both operation procedures tested in this work. (**a**) Mixing and homogenization of rhodamine B solution and water, and (**b**) MSA-DLLME of LMG-Al complex with n-hexanol [27]. ↑ ↓ dispense.

**Figure 6 molecules-27-07279-f006:**
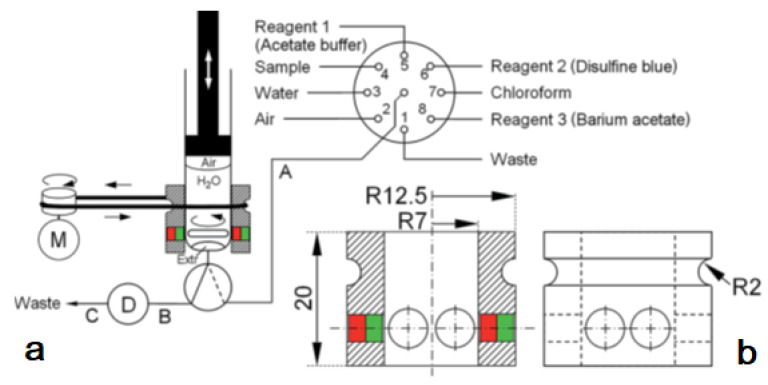
(**a**): analyzer manifold with the selection valve (SV), syringe (S), solenoid 3-way head valve (V), detection flow cell (D), and DC motor (M). PTFE tubing (0.8 mm id) A: 35 cm, B: 10 cm, and C: 40 cm.; (**b**): the magnetic stirring bar driver design given in detail consisting of a Delrin^®®^ tube and two neodymium magnets [37].

**Figure 7 molecules-27-07279-f007:**
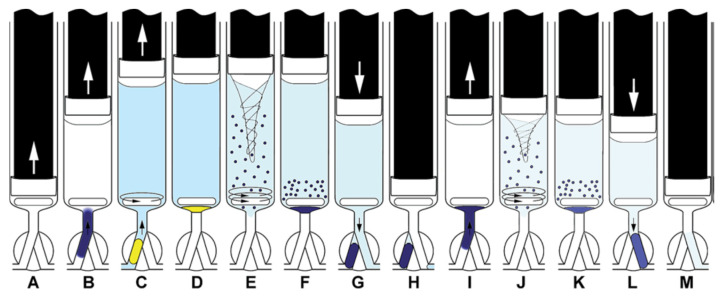
Operation scheme of extraction with simple extract washing. Aspiration of the sample, buffer, and DSB (**A**,**B**), mixing (**C**) and aspiration of the extracting solvent (ExtrS) and air (**D**), MSA-DLLME. (**E**). Sedimentation of ExtrS (**F**), saving ExtrS in HC and discharge of the aqueous phase to waste (**G**,**H**), aspiration of DSB, barium acetate, and water (**I**), washing of ExtrS by MSA-DLLME (**J**), sedimentation of ExtrS (**K**), propelling ExtrS to detector (**L**), syringe content discharge to waste (**M**) [37]. ↑ ↓ dispense.

**Figure 8 molecules-27-07279-f008:**
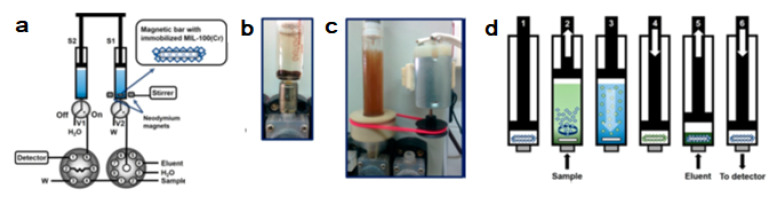
(**a**) Scheme of the flow-based instrumental setup. V1–V2, solenoid valves; S1–S2, 5 mL glass syringes; W, waste reservoir; (**b**) Magnet coated with 10 mg of magnetic MOF placed inside a 5 mL glass syringe; (**c**) In-syringe dispersion of a magnetic MOF facilitated by magnetic stirring; (**d**) Schematic depiction of the extraction procedure in syringe 1 (washing step between extraction and elution is not included) [38].

**Figure 9 molecules-27-07279-f009:**
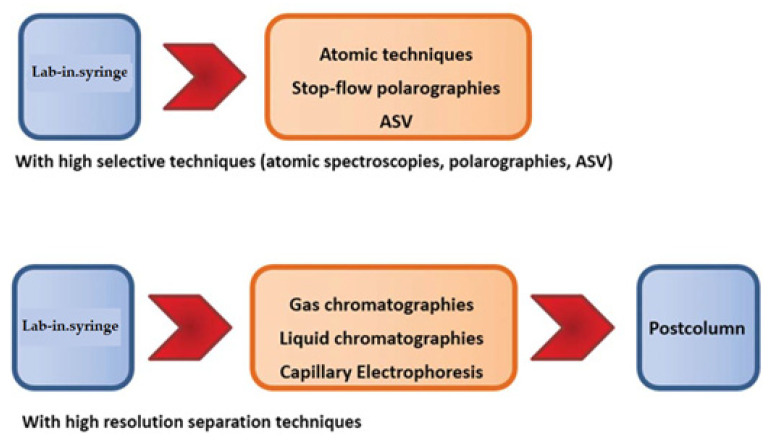
Hyphenation possibilities of LIS with atomic spectroscopies (spectrometries), chromatographies and using stop-flow techniques.

**Figure 10 molecules-27-07279-f010:**
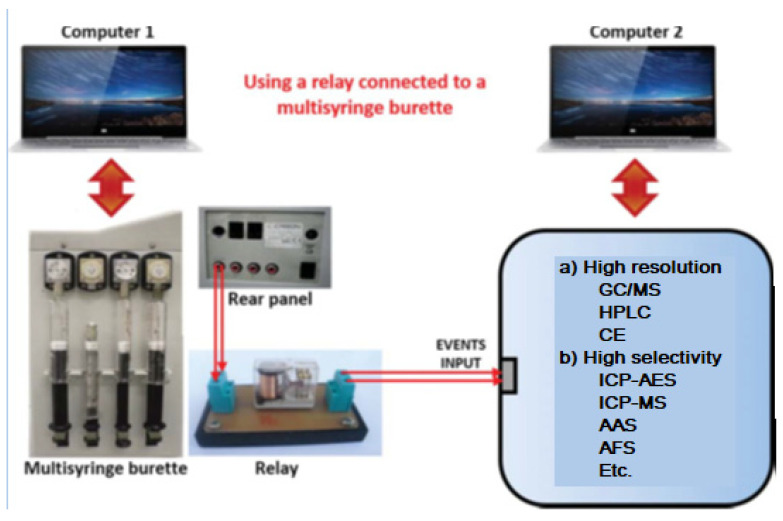
Coupling a multisyringe burette using a relay as an interface connected to the event input of another instrument.

**Figure 11 molecules-27-07279-f011:**
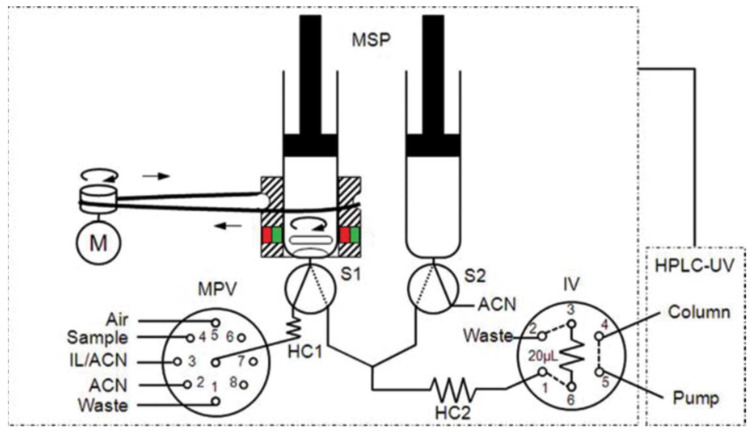
Schematic manifold of the in-syringe magnetic stirring-assisted dispersive liquid–liquid microextraction system coupled to HPLC (in-syringe MSA–DLLME–HPLC) for UV filters determination using UV detection. The manifold was composed of a multisyringe pump (MSP) with two syringes (S1 and S2) with a magnetic stirring system on S1, a multiposition valve (MPV), a high injection valve (IV), a DC motor (M) and an HPLC system [47].

**Figure 12 molecules-27-07279-f012:**
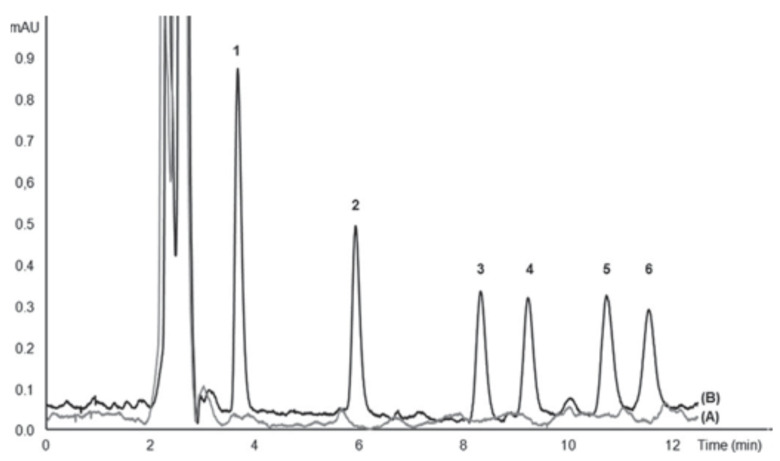
Chromatogram of UV filters analyzed with the proposed in-syringe MSA-DLLME-HPLC system: (A) surface seawater and (B) spiked surface seawater with 25 mg L^−1^ 2-Hydroxy-4-methoxybenzophenone (1), 10 mg L^−1^ 3-(4 methylbenzylidene) camphor (2), 25 mg L^−1^ cyano-3,3-diphenylacrylate (3), 10 mg L^−1^ 2-ethylhexyl 4-dimethylaminobenzoate (4), 60 mg L^−1^ ethylhexyl salicylate (5), 60 mg L^−1^ homosalate (6) [47].

**Figure 13 molecules-27-07279-f013:**
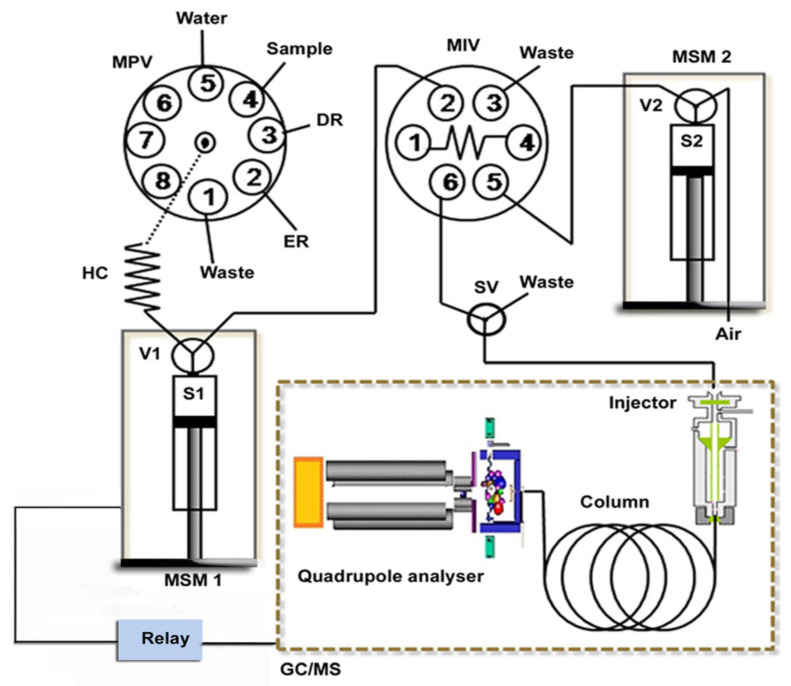
Schematic illustration of the in-syringe-DLLME–GC/MS setup for on-line preconcentration and determination of phthalates. DR: dispersive reagent (acetone), ER: extraction reagent (trichloroethylene), HC: holding coil, MIV: microinjection valve, MPV: multiposition valve, MSM1–2: multisyringe module 1–2, S1–2: syringe pump 1–2, SV: solenoid valve and V1–2: valve 1–2 [48].

**Figure 14 molecules-27-07279-f014:**
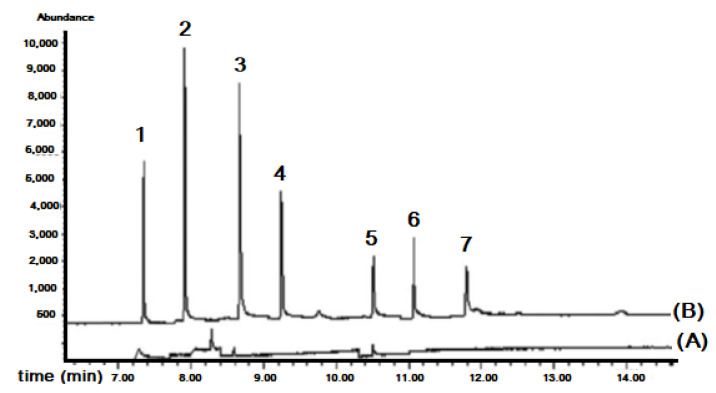
Chromatograms of tap water analyzed with the in-syringe-DLLME–GC/MS: (A) non-spiked and (B) spiked with 5 μg L^−1^ of studied phenyl phthalates. Peaks numbers correspond to: (1) dimethyl phthalate, (2) diethyl phthalate, (3) benzyl benzoate (ICS), (4) dibutyl phthalate, (5) butylbenzyl phthalate, (6) diethylhexyl phthalate, (7) di-n-octyl phthalate [48].

**Figure 15 molecules-27-07279-f015:**
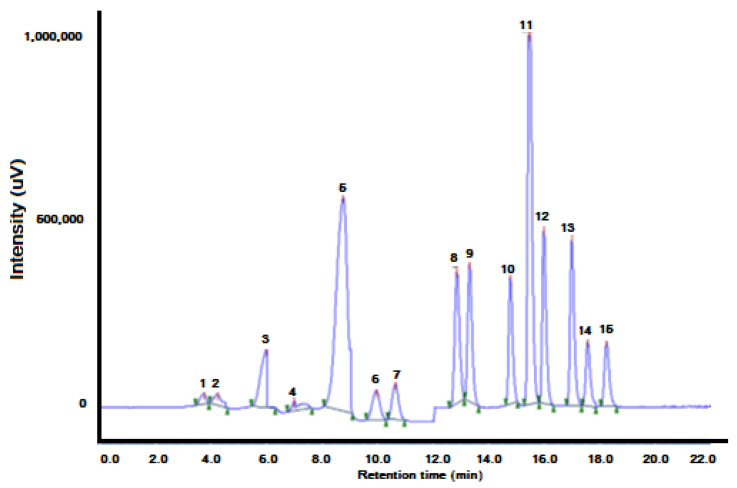
On-line DLLME–HPLC chromatogram of water spiked with 0.02 μg L^−1^ of each PAH. PAHs were preconcentrated from 4 mL of sample. The elution order and retention times are shown in Table 3. (1) Naphthalene (Nap), (2) acenaphthene (Acp), (3) fluorene (Flu), (4) phenanthrene (PA), (5) anthracene (Ant), (6) fluoranthene (FL), (7) pyrene (Pyr), (8) benz[a]anthracene (BaA), (9) chrysene (Chr), (10) benzo[b]fluoranthene (BbFl), (11) benzo[k]fluor- anthene (BkFl), (12) benzo[a]pyrene (BaP), (13) dibenzo[a,h]anthracene (DBA), (14) benzo[g,h,i]perylene (BghiP) and (15) indeno [1,2,3-cd]pyrene (IP) [60].

**Table 1 molecules-27-07279-t001:** Comparison of several methods for aluminum determination using solvent emulsification or DLLME.

Volume of Extraction Solvent μL	RSD %	LOD ppb	URL ppb	Sample mL	Time min	Extraction	Detection	Ref.
600	1.7	0.05	−15	25	<10	IL-DLLME	FL	[32]
132	4.5	0.8	−250	20	>8	DLLME-SFO	ICP-OES	[33]
48	2.6–5.3	0.6–0.9	−1000	10	>11	US-DLLME	ICP-OES	[34]
75	3.2	1.7	n.g.	10	>15	USILDLLME	UV-Vis	[35]
98	1.87	0.13	−1000	10	>10	US-DLLME	ICP-OES	[36]
950	<5	0.22	−27	3.9	4.4	In-syringe DSA-DLLME	FL	[28]
150	3.3–4.4	0.16	−33	4.1	3.5	In-syringe DSA-DLLME	FL	[27]

Abbreviations: DSA, dispersion solvent assisted; DLLME, dispersive liquid–liquid micro-extraction; FL, fluorescence; IL, ionic-liquid based; IS-MSA, in-syringe magnetic stirring assisted; n.g., not given; SE, surfactant enhanced; SFO, solidification of organic drop; US, ultrasound assisted; ULR, upper linear working range limit; UV–vis, spectrophotometry.

**Table 2 molecules-27-07279-t002:** Several LIS methods hyphenated with chromatographies.

Sample	RSD %	LOD	Working Range	Method	Detection	Ref.
UV filters	6 & 8	0.08 & 12 μg L^−1^	up to 500 μg L^−1^	IS-MSA	HPLC	[47]
Nonsteroidal Anti-Inflammatory Drugs	3.2 & 7.6	0.06 & 1.98 µg L^−1^	up to 200 µg L^−1^	IS-MSA	HPLC	[49]
Volatile fatty acids	0.7	0.1 & 1.3 mg L^−1^	up to 1000 mg L^−1^	IS-MSA	HPLC	[50]
Sulfonamide antibiotics from urine	≤5	7.5 μg L^−1^	50–5000 μg/L	IS-MSA	HPLC	[51]
fluoroquinolones	<3	20 ng L^−1^ to 30 ng L^−1^	n.g.	SDE		[52]
Priority phenolic pollutants	4.4	40 μg L^−1^	20,000 μg L^−1^	IS-MSA	MSC	[53]
Estrogens in wastewater	≤7.06	112 ng L^−1^	up to 50,000 ng L^−1^	IS-MSA	GC	[54]
Four hydrocarbons waters	<4	1–2 μg L^−1^	0.016–1 mg L^−1^	HS	GC	[55]
Estrogens	<6	11 ng L^−1^	Up to 1000 μg L^−^^1^	MC	GC/MS	[56]
Six phthalates in water	<5	0.03 & 0.10 g L^−1^	0.5–120 μg L^−1^	DLLME	GC/MS	[48]
Herbicides in waters	6.6 & 7.4	0.045 & 0.03 μg L^−1^	up to 200 μg L^−1^	IS-MSA	GC/MS	[57]
Long-chain fatty acids	≤7.9	0.01 & 0.05 mg L^−1^	up to 100 mg L^−1^	DLLME	GC/MS	[58]
Ultraviolet filters in water	5.5 & 17	0.023 & 0.16 μg L^−1^	up to 500 μg L^−1^	IS-MSA	GC/MS	[59]
polycyclic aromatic hydrocarbons	1.6–4	0.02–0.6 μg L^−1^	0.2–600 μg L^−1^	DLLME	HPLC	[60]

Abbreviations: MSC, Multisyringe chromatography; DLLME, dispersive liquid–liquid mixroextraction; IS-MSA, in-syringe magnetic stirring assisted; SDE, ingle-drop microextraction; HS, air for headspace formation; MC, magnetic carbons; n.g., not given; ULR, upper linear working range limit.

**Table 3 molecules-27-07279-t003:** PAHs: Retention times, variation coefficient, enrichment factor, reproducibility of the method, limits of detection.

Compound	Retention Order	t_ret_ (min)	R^2^	EF	RSD% (*n* = 5)		LOQ (μg L^−1^)
					Inter Day	Intraday	
Nap	1	4.2	0.9989	87.56	4.69	4.34	0.61
Acp	2	6.0	0.9997	86.93	4.13	3.82	0.52
Flu	3	6.2	0.9999	92.06	3.84	2.61	0.09
PA	4	7.4	0.9998	92.36	4.72	5.31	0.16
Ant	5	8.7	0.9997	95.61	1.67	2.89	0.04
FL	6	10.0	0.9997	86.08	4.23	4.73	0.41
Pyr	7	10.7	0.9999	87.62	3.74	4.51	0.26
BaA	8	12.9	0.9995	90.51	3.48	3.86	0.08
Chr	9	13.3	0.9998	88.05	3.01	2.95	0.07
BbFl	10	14.8	0.9997	89.78	2.58	3.74	0.08
BkFl	11	15.4	0.9999	95.45	1.63	2.13	0.02
BaP	12	16.0	0.9999	90.87	2.35	3.46	0.05
DBA	13	17.5	0.9999	88.35	4.37	3.86	0.14
BghiP	14	18.1	0.9997	91.83	2.96	3.59	0.09
IP	15	16.9	0.9998	91.24	2.85	2.97	0.06

**Table 4 molecules-27-07279-t004:** PAHs content and recoveries for three different kinds of spiked waters.

Analytes	Tap Water		Rain Water		Stream Surface Waters	
	Content	RR% ± RSD	Content	RR% ± RSD	Content	RR% ± RSD
Nap	nd	93 ± 4	0.38	89 ± 4	0.41	92 ± 3
Acp	nd	96 ± 3	nd	97 ± 2	0.13	96 ± 2
Flu	nd	98 ± 2	nd	95 ± 3	nd	97 ± 3
PA	nd	93 ± 2	0.43	98 ± 3	0.35	95 ± 2
Ant	nd	98 ± 2	0.46	106 ± 3	0.52	99 ± 3
FL	nd	96 ± 4	0.51	96 ± 3	0.29	94 ± 2
Pyr	nd	94 ± 2	0.42	95 ± 3	0.29	97 ± 1
BaA	nd	97 ± 3	nd	98 ± 2	nd	96 ± 3
Chr	nd	98 ± 2	nd	96 ± 2	0.48	99 ± 2
BbFl	nd	96 ± 3	nd	97 ± 2	nd	100 ± 4
BkFl	nd	102 ± 2	nd	99 ± 4	nd	97 ± 3
BaP	nd	97 ± 4	nd	95 ± 3	nd	98 ± 2
DBA	nd	99 ± 1	nd	102 ± 3	nd	103 ± 2
BghiP	nd	97 ± 4	nd	94 ± 3	nd	96 ± 3
IP	nd	92 ± 3	nd	95 ± 2	0.31	96 ± 2

nd: not detected; RR:recoveries; Content in μg L^−1^.

## Data Availability

Not appliable.

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
