# Peer review of "Lab-in-Syringe, a Useful Technique for the Analysis and Detection of Pollutants of Emerging Concern in Environmental and Food Samples"

_molecules, 2022, doi:10.3390/molecules27217279_

Round 1

Reviewer 1 Report

A review should not only list what others have done but also contain thoughts and ideas of the authors. And it must be legible. Already the first two pages contain so many language problems that one does not want to read further. Figures are lifted of other publication without stating it. Abstract and Introduction are painfully short. This all has the look-and-feel of a quick way to get citations.

As the manuscript is it must be rejected.

Author Response

Reviewer 1

Comments and Suggestions for Authors

A review should not only list what others have done but also contain thoughts and ideas of the authors. And it must be legible. Already the first two pages contain so many language problems that one does not want to read further. Figures are lifted of other publication without stating it. Abstract and Introduction are painfully short. This all has the look-and-feel of a quick way to get citations.

As the manuscript is it must be rejected.

The opinion of referees 2 and 3 contrasts and contradicts a lot with that of referee 1

Referee 2 opinion is:

The submitted article ‘Lab-in-syringe, a useful technique for the Analysis and Detection of Pollutants of Emerging Concern in Environmental and Food Samples’ is very interesting and professional article. This is a kind of rare studies, hence is very desire. It should be emphasized that Lab-in-syringe (LIS) is a recently conceived liquid-liquid and liquid-solid extraction technique that can be very useful for the preconcentration of analytes and elimination its interferences, however it is difficult task.. In my opinion manuscript is written correctly, concisely in most parts (some mistakes and remarks should be corrected before acceptance for publication). Below I pointed most of mistakes and matters for explanation.

            The mistakes have been corrected according with the remarks of this referee

Referee 3 opinion is:

The manuscript was written in a clear and concise language and the LIS analytical methodology is explained in detail.

There are very small issues regarding English, as the examples bellow:

            The mistakes have been corrected according with the remarks of this referee

Reviewer 2 Report

The submitted article ‘Lab-in-syringe, a useful technique for the Analysis and Detection of Pollutants of Emerging Concern in Environmental and Food Samples’ is very interesting and professional article. This is a kind of rare studies, hence is very desire. It should be emphasized that Lab-in-syringe (LIS) is a recently conceived liquid-liquid and liquid-solid extraction technique that can be very useful for the preconcentration of analytes and elimination its interferences, however it is difficult task.. In my opinion manuscript is written correctly, concisely in most parts (some mistakes and remarks should be corrected before acceptance for publication). Below I pointed most of mistakes and matters for explanation.

1.      In introduction the emphasis should be placed on justification

2.      More citation should be included, especially for related studies and topics:

Alachkar, A., Alhassan, S., & Senel, M. (2022). Lab-in-a-syringe: A novel electrochemical biosensor for on-site and real-time monitoring of dopamine in freely behaving mice. ACS sensors, 7(1), 331-337.

3.      The good idea should be summary as graphical workflow in introduction

4.      Conclusion part should be more informative (more details, perhaps in brackets?)

5.      Please include advances and disadvantages of Your studies in conclusions

I totally agree that this is very important subject, and it is very important for publication in MDPI. I will recommend this article for publication in MOLECULES after minor revision.

Author Response

Reviewer 2

Comments and Suggestions for Authors

The submitted article ‘Lab-in-syringe, a useful technique for the Analysis and Detection of Pollutants of Emerging Concern in Environmental and Food Samples’ is very interesting and professional article. This is a kind of rare studies, hence is very desire. It should be emphasized that Lab-in-syringe (LIS) is a recently conceived liquid-liquid and liquid-solid extraction technique that can be very useful for the preconcentration of analytes and elimination its interferences, however it is difficult task.. In my opinion manuscript is written correctly, concisely in most parts (some mistakes and remarks should be corrected before acceptance for publication). Below I pointed most of mistakes and matters for explanation.

  1. In introduction the emphasis should be placed on justification

Done at the end of the introduction

  1. More citation should be included, especially for related studies and topics:

Alachkar, A., Alhassan, S., & Senel, M. (2022). Lab-in-a-syringe: A novel electrochemical biosensor for on-site and real-time monitoring of dopamine in freely behaving mice. ACS sensors, 7(1), 331-337.

The proposed reference describes a manual system using a μSyringe, which do not have any relationship with the lab-in-syringe proposed in this paper

  1. The good idea should be summary as graphical workflow in introduction

I don’t know how to do it

  1. Conclusion part should be more informative (more details, perhaps in brackets?)

                  Done

  1. Please include advances and disadvantages of Your studies in conclusions

                  Done

I totally agree that this is very important subject, and it is very important for publication in MDPI. I will recommend this article for publication in MOLECULES after minor revision.

Reviewer 3 Report

General comments:

The manuscript was written in a clear and concise language and the LIS analytical methodology is explained in detail.

There are very small issues regarding English, as the examples bellow:

-          Line 12 - preconcentration of analytes and elimination OF its interferences

-          Line 22 - The Dispersive Liquid-Liquid Microextraction (DLLME) technique [1] WAS developed by Rezaee et al.

I detected some confusion between the English abbreviation from Gas-Chromatography Mass Spectrometry (GC/MS) and the corresponding Portuguese abbreviation CG/MS. Please correct Figure 10 and lines 543 and 544. In addition, authors must choose between using GC/MS or GC-MS and use only one abbreviation. Please revise the manuscript and harmonize the abbreviations.

-          Legend of Figure 2.c): “Chromatograms corresponding to the direct injection of tap water spiked with 15 µg L−1 benzo(a)pyrene spiked tap water after DLLME, and a standard containing 15 µg L−1 of benzo(a)pyrene using DLLME.”

A comma is missing after the first benzo(a)pyrene.

-          Figure 4: The “A” and “B” from the two subfigures is mixed with the “A” and “B” used within the Figure. Authors should follow the same notation that was adopted to the previous figures, as: Figure 4 a) and Figure 4b). Please revise the notation used on the remaining figures.

-          Lines 549 and 550 – There´s a problem with the use of letters with an accent mark: Conselho Nacional de Desenvolvimento Científico e Tecnológico… Fundação de Amparo à Pesquisa…

Line 466 - Hyphenated technique of LIS with chromatographic techniques

Authors present LIS as able to be hyphenated with “big instruments” as ICP–MS and chromatographic techniques as GC/MS or LC/MS. They also present an example of its use to quantify UV filters in surface seawater and swimming pool samples. However, they do not no exhibit chromatograms of those samples.

The application of LIS to real samples is the crucial step necessary to validate the methodology, and therefore should be presented in detail (with chromatograms).  Although useful, figure 12 has a lower impact on the validation of the methodology, as authors used spiked surface seawater and not real samples. Additionally, authors should comment the concentration levels used and compare it with values usually found at surface seawaters.

Author Response

Reviewer 3

Comments and Suggestions for Authors

General comments:

The manuscript was written in a clear and concise language and the LIS analytical methodology is explained in detail.

There are very small issues regarding English, as the examples bellow:

-          Line 12 - preconcentration of analytes and elimination OF its interferences

                  Done

-         Line 22 - The Dispersive Liquid-Liquid Microextraction (DLLME) technique [1] WAS developed by Rezaee et al.

                  Done

I detected some confusion between the English abbreviation from Gas-Chromatography Mass Spectrometry (GC/MS) and the corresponding Portuguese abbreviation CG/MS. Please correct Figure 10 and lines 543 and 544. In addition, authors must choose between using GC/MS or GC-MS and use only one abbreviation. Please revise the manuscript and harmonize the abbreviations.

            GC/MS has been selected along the paper

-         Legend of Figure 2.c): “Chromatograms corresponding to the direct injection of tap water spiked with 15 µg L−1 benzo(a)pyrene spiked tap water after DLLME, and a standard containing 15 µg L−1 of benzo(a)pyrene using DLLME.”

A comma is missing after the first benzo(a)pyrene.

            Done

-         Figure 4: The “A” and “B” from the two subfigures is mixed with the “A” and “B” used within the Figure. Authors should follow the same notation that was adopted to the previous figures, as: Figure 4 a) and Figure 4b). Please revise the notation used on the remaining figures.

                  Done, revised all figures

-         Lines 549 and 550 – There´s a problem with the use of letters with an accent mark: …Conselho Nacional de Desenvolvimento Científico e Tecnológico… Fundação de Amparo à Pesquisa…

                  Done

Line 466 - Hyphenated technique of LIS with chromatographic techniques

Authors present LIS as able to be hyphenated with “big instruments” as ICP–MS and chromatographic techniques as GC/MS or LC/MS. They also present an example of its use to quantify UV filters in surface seawater and swimming pool samples. However, they do not no exhibit chromatograms of those samples.

            Figure 12 are chromatograms of surface of spiked and non-spiked seawater

            Figure 14 are chromatograms of several phthalates in non-spiked and spiked tap waters

            In references 51, 57 and 60 more other examples might be found

  • The application of LIS to real samples is the crucial step necessary to validate the methodology, and therefore should be presented in detail (with chromatograms).  Although useful, figure 12 has a lower impact on the validation of the methodology, as authors used spiked surface seawater and not real samples. Additionally, authors should comment the concentration levels used and compare it with values usually found at surface seawaters.

Figures 12, 14 and 15 correspond to the chromatograms of spiked and non-spiked real samples

Round 2

Reviewer 1 Report

None of my complaints of the first review cycle have been addressed by the authors. To just say it is contrary to the other two reviewers is insufficient. There is a reason why more than one reviewer is invited!

The abstract consists of just two sentences. Is a review article not worth a bit more effort? In one of these two sentences the authors mention "liquid-solid extraction" but in the also painfully short Introduction only DLLME is mentioned. And the English has not improved at all. The three sentences in the Introduction starting at line 27 are just not proper English. And it does not stop there. There a numerous locations with grammatical problems and sentences which do not make sense.

I do not know what the publication policy of Molecules is but if a graphic were lifted out of a publication I authored, open access or not, I would expect that that graphic is properly referenced! None of he graphics in this manuscript show any citations.

I also miss a critical reflection of the techniques presented here. It took the authors about two thirds of the manuscript until they mention that the selectivity of the presented methodology is insufficient for true analysis of environmental contaminants. And then they come with section 2.5.1. and figures 9 & 10 which are so trivial they should not be included in this manuscript.

The Conclusions could also be a bit more elaborate. Last but not least, I do not see how there is no conflict of interest from the authors. At least the lead author's affiliation points to a company dealing with equipment necessary for LIS. That would make a conflict in my books.

I am sorry but I have to stand to my initial judgement that this manuscript needs to be rejected in its current state.

Author Response

The abstract consists of just two sentences. Is a review article not worth a bit more effort? In one of these two sentences the authors mention "liquid-solid extraction" but in the also painfully short Introduction only DLLME is mentioned. And the English has not improved at all.

            The abstract content has been changed

The three sentences in the Introduction starting at line 27 are just not proper English. And it does not stop there. There a numerous locations with grammatical problems and sentences which do not make sense.

            These sentences have been modified

I do not know what the publication policy of Molecules is but if a graphic were lifted out of a publication I authored open access or not, I would expect that that graphic is properly referenced! None of the graphics in this manuscript show any citations.

The graphics citations are included in the text, and also included in the figures

I also miss a critical reflection of the techniques presented here. It took the authors about two thirds of the manuscript until they mention that the selectivity of the presented methodology is insufficient for true analysis of environmental contaminants. And then they come with section 2.5.1. and figures 9 & 10 which are so trivial they should not be included in this manuscript.

In our opinion, figures 9 & 10 may be very helpful for people who are not very familiar with this technique

We try to say that the presented methodology may be very useful for the analysis of many environmental contaminants, but not for all

The Conclusions could also be a bit more elaborate. Last but not least, I do not see how there is no conflict of interest from the authors. At least the lead author's affiliation points to a company dealing with equipment necessary for LIS. That would make a conflict in my books.

Sciware Systems was a Spin Off of the University of the Balearic Islands created by one of the authors. Currently this authors is retired and the company has been closed.